# DRUM: End-To-End Differentiable Rule Mining On Knowledge Graphs

**Ali Sadeghian** [*1], **Mohammadreza Armandpour**[*2], **Patrick Ding,**[2] **Daisy Zhe Wang,**[1]

[1] Department of Computer Science, University of Florida
[2] Department of Statistics, Texas A&M University
`{asadeghian, daisyw}@ufl.edu, {armand, patrickding}@stat.tamu.edu`

## Abstract

In this paper, we study the problem of learning probabilistic logical rules for inductive and interpretable link prediction. Despite the importance of inductive link prediction, most previous works focused on transductive link prediction and cannot manage previously unseen entities. Moreover, they are black-box models that are not easily explainable for humans. We propose DRUM, a scalable and differentiable approach for mining first-order logical rules from knowledge graphs which resolves these problems. We motivate our method by making a connection between learning confidence scores for each rule and low-rank tensor approximation. DRUM uses bidirectional RNNs to share useful information across the tasks of learning rules for different relations. We also empirically demonstrate the efficiency of DRUM over existing rule mining methods for inductive link prediction on a variety of benchmark datasets.

## 1   Introduction

Knowledge bases store structured information about real-world people, locations, companies and governments, etc. Knowledge base construction has attracted the attention of researchers, foundations, industry, and governments [11, 13, 34, 38]. Nevertheless, even the largest knowledge bases remain incomplete due to the limitations of human knowledge, web corpora, and extraction algorithms.

Numerous projects have been developed to shorten the gap between KBs and human knowledge. A popular approach is to use the existing elements in the knowledge graph to infer the existence of new ones. There are two prominent directions in this line of research: representation learning that obtains distributed vectors for all entities and relations in the knowledge graph [12, 31, 33], and rule mining that uses observed co-occurrences of frequent patterns in the knowledge graph to determine logical rules [5, 15]. An example of knowledge graph completion with logical rules is shown in Figure 1.

One of the main advantages of logic-learning based methods for link prediction is that they can be applied to both transductive and inductive problems while representation learning methods like that of Bordes et al. [4] and Yang et al. [40] cannot be employed in inductive scenarios. Consider the scenario in Figure 1, and suppose that at training time our knowledge base does not contain information about Obama's family. Representation learning techniques need to be retrained on the whole knowledge base in order to find the answer. In contrast rule mining methods can transfer reasoning to unseen facts.

Additionally, learning logical rules provides us with interpretable reasoning for predictions which is not the case for the embedding based method. This interpretability can keep humans in the loop, facilitate debugging, and increase user trustworthiness. More importantly,

---

rules allow domain knowledge transfer by enabling the addition of extra rules by experts, a strong advantage over representation learning models in scenarios with little or low-quality data.

Mining rules have traditionally relied on pre-defined statistical measures such as support and confidence to assess the quality of rules. These are fixed heuristic measures, and are not optimal for various use cases in which one might want to use the rules. For example, using standard confidence is not necessarily optimal for statistical relational learning. Therefore, finding a method that is able to simultaneously learn rule structures as well as appropriate scores is crucial. However, this is a challenging task because the method needs to find an optimal structure in a large discrete space and simultaneously learn

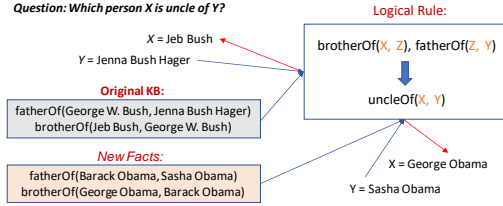

Figure 1: Using logical rules for knowledge base reasoning

proper score values in a continuous space. Most previous approaches address parts of this problem [9, 20, 22, 39] but are not able to learn both structure and scores together, with the exception of Yang et al. [41].

In this paper we propose DRUM, a fully differentiable model to learn logical rules and their related confidence scores. DRUM has significant importance because it not only addresses the aforementioned challenges, but also allows gradient based optimization to be employed for inductive logic programming tasks.

Our contributions can be summarized as: 1) An end-to-end differentiable rule mining model that is able to learn rule structures and scores simultaneously; 2) We provide a connection between tensor completion and the estimation of confidence scores; 3) We theoretically show that our formulation is expressive enough at finding the rule structures and their related confidences; 4) Finally, we demonstrate that our method outperforms previous models on benchmark knowledge bases, both on the link prediction task and in terms of rule quality.

## 2 Problem Statement

**Definitions**     We model a knowledge graph as a collection of facts $G = \{(s, r, o)|s, o \in \mathcal{E}, r \in \mathcal{R}\}$, where $\mathcal{E}$ and $\mathcal{R}$ represent the set of entities and relations in the knowledge graph, respectively.

A first order logical rule is of the form $\mathbf{B} \Longrightarrow H$, where $\mathbf{B} = \bigwedge_i B_i(\cdots)$ is a conjunction of atoms $B_i$, e.g., $livesIn(\cdots)$, called the *Body*, and $H$ is a specific predicate called the head. A rule is *connected* if every atom in the rule shares at least one variable with another atom, and a rule is *closed* if each variable in the rule appears in at least two atoms.

**Rule Mining**     We address the problem of learning first-order logical Horn clauses from a knowledge graph. In particular we are interested in mining closed and connected rules. These assumptions ensure finding meaningful rules that are human understandable. Connectedness also prevents finding rules with unrelated relations.

Formally, we aim to find all $T \in \mathbb{N}$ and relations $B_1, B_2, \cdots, B_T, H$ as well as a *confidence value* $\alpha \in \mathbb{R}$, where:

$$B_1(x, z_1) \wedge B_2(z_1, z_2) \wedge \cdots \wedge B_T(z_{T-1}, y) \implies H(x, y) \ : \alpha, \tag{1}$$

where, $z_i$s are variables that can be substituted with entities. This requires searching a discrete space to find $B_i$s and searching a continuous space to learn $\alpha$ for every particular rule.

## 3 Related work

Mining Horn clauses has been previously studied in the Inductive Logic Programming (ILP) field, e.g, FOIL [29], MDIE [26] and Inspire [32]. Given a background knowledge base, ILP provides a framework for learning on multi-relational data. However, despite the strong representation powers of ILP, it requires both positive and negative examples and does not scale to large datasets. This is a huge drawback since most knowledge bases are large and contain only positive facts.

Recent rule mining methods such as AMIE+ [15] and Ontological Pathfinding (OP) [5] use predefined metrics such as confidence and support and take advantage of various parallelization and partitioning techniques to speed up the counting process. However, they still suffer from the inherent limitations of relying on predefined confidence and discrete counting.

Most recent knowledge base rule mining approaches fall under the same category as ILP and OP. However, Yang et al. [40] show that one can also use graph embeddings to mine rules. They introduce DistMult, a simple bilinear model for learning entity and relation representations. The relation representations learned via the bilinear model can capture compositional relational semantics via matrix multiplications. For example, if the rule $B_1(x,y) \land B_2(y,z) \implies H(x,z)$ holds, then intuitively so should $\mathbf{A}_{B_1}\mathbf{A}_{B_2} \approx \mathbf{A}_H$. To mine rules, they use the Frobenius norm to search for all possible pairs of relations with respect to their compositional relevance to each head. In a more recent approach Omran et al. [28] improve this method by leveraging pruning techniques and computing traditional metrics to scale it up to larger knowledge bases.

In [16] the authors proposed a RESCAL-based model to learn from paths in KGs. More recently, Yang et al. [41] provide the first fully differentiable rule mining method based on TensorLog [6], Neural LP. They estimate the graph structure via constructing TensorLog operators per relation using a portion of the knowledge graph. Similar to us, they chain these operators to compute a score for each triplet, and learn rules by maximizing this score. As we explain in Section 4.1, this formulation is bounded to a fixed length of rules. To overcome this limitation, Neural LP uses an LSTM and attention mechanisms to learn variable rule lengths. However, it can be implied from Theorem 1 that its formulation has theoretical limitations on the rules it can produce.

There are some other interesting works [7, 14, 25, 30] which learn rules in a differentiable manner. However, they need to learn embeddings for each entity in the graph and they do link prediction not only based on the learned rules but also the embeddings. Therefore we exclude them from our experiment section.

# 4 Methodology

To provide intuition about each part of our algorithm we start with a vanilla solution to the problem. We then explain the drawbacks of the this approach and modify the suggested method step-by-step to makes the challenges of the problem more clear and provides insight into different parts of the suggested algorithm.

We begin by defining a one-to-one correspondence between the elements of $\mathcal{E}$ and $\{\mathbf{v}_1, ..., \mathbf{v}_n\}$, where $n$ is the number of entities and $\mathbf{v}_i \in \{0,1\}^n$ is a vector with 1 at position $i$ and 0 otherwise. We also define $\mathbf{A}_{B_r}$ as the adjacency matrix of the knowledge base with respect to relation $B_r$; the $(i,j)^{\text{th}}$ elements of $\mathbf{A}_{B_r}$ equals to 1 when the entities corresponding to $\mathbf{v}_i$ and $\mathbf{v}_j$ have relation $B_r$, and 0 otherwise.

## 4.1 A Compact Differentiable Formulation

To approach this inherently discrete problem in a differentiable manner, we utilize the fact that using the above notations for a pair of entities $(x,y)$ the existence of a chain of atoms such as

$$B_1(x,z_1) \land B_2(z_1,z_2) \land \cdots \land B_T(z_{T-1},y) \tag{2}$$

is equivalent to $\mathbf{v}_x^T \cdot \mathbf{A}_{B_1} \cdot \mathbf{A}_{B_2} \cdots \mathbf{A}_{B_T} \cdot \mathbf{v}_y$ being a positive scalar. This scalar is equal to the number of paths of length $T$ connecting $x$ to $y$ which traverse relation $B_{r_i}$ at step $i$. It is straightforward to show that for each head relation $H$, one can learn logical rules by finding an appropriate $\boldsymbol{\alpha}$ in

$$\omega_H(\boldsymbol{\alpha}) \doteq \sum_s \alpha_s \prod_{k \in p_s} \mathbf{A}_{B_k} \tag{3}$$

that maximizes

$$O_H(\boldsymbol{\alpha}) \doteq \sum_{(x,H,y) \in KG} \mathbf{v}_x^T \omega_H(\boldsymbol{\alpha}) \mathbf{v}_y, \tag{4}$$

where $s$ indexes over all potential rules with maximum length of $T$, and $p_s$ is the ordered list of relations related to the rule indexed by $s$.

However, since the number of learnable parameters in $O_H(\boldsymbol{\alpha})$ can be exceedingly large, i.e. $\mathcal{O}(|\mathcal{R}|^T)$, and the number of observed pairs $(x, y)$ which satisfy the head $H$ are usually small, direct optimization of $O_H(\boldsymbol{\alpha})$ falls in the regime of over-parameterization and cannot provide useful results. To reduce the number of parameters one can rewrite $\omega_H(\boldsymbol{\alpha})$ as

$$\Omega_H(\mathbf{a}) \doteq \prod_{i=1}^{T} \sum_{k=1}^{|\mathcal{R}|} a_{i,k} \mathbf{A}_{B_k}. \tag{5}$$

This reformulation significantly reduces the number of parameters to $T\mathcal{R}$. However, the new formulation can only learn rules with fixed length $T$. To overcome this problem, we propose to modify $\Omega_H(\mathbf{a})$ to

$$\Omega_H^I(\mathbf{a}) \doteq \prod_{i=1}^{T} (\sum_{k=0}^{|\mathcal{R}|} a_{i,k} \mathbf{A}_{B_k}), \tag{6}$$

where we define a new relation $B_0$ with an identity adjacency matrix $A_{B_0} = I_n$. With this change, the expansion of $\Omega_H^I$ includes all possible rule templates of length $T$ or smaller with only $T(|\mathcal{R}| + 1)$ free parameters.

Although $\Omega_H^I$ considers all possible rules lengths, it is still constrained in learning the correct rule confidences. As we will show in the experiments Section 5.3, this formulation (as well as Neural LP [41]) inevitably mines incorrect rules with high confidences. The following theorem provides insight about the restricted expressive power of the rules obtained by $\Omega_H^I$.

**Theorem 1.** *If $R_o$, $R_s$ are two rules of length $T$ obtained by optimizing the objective related to $\Omega_H^I$, with confidence values $\alpha_o, \alpha_s$, then there exists $\ell$ rules of length $T$, $R_1, R_2, \cdots, R_\ell$, with confidence values $\alpha_1, \alpha_2, \cdots, \alpha_\ell$ such that:*

$$d(R_o, R_1) = d(R_\ell, R_s) = 1 \quad and \quad d(R_o, R_s) \le \ell + 1,$$
$$d(R_l, R_{l+1}) = 1 \quad and \quad \alpha_l \ge \min(\alpha_o, \alpha_s) \quad for \quad 1 \le l \le \ell,$$

*where $d(.,.)$ is a distance between two rules of the same size defined as the number of mismatched atoms in their bodies.*

*Proof.* The proof is provided in the supplementary file. $\qquad\square$

To further explain Theorem 1, consider an example knowledge base with only two meaningful logical rules of body length $T = 3$, i.e. $R_o$ and $R_s$ such that they do not share any body atoms. According to Theorem 1, learning these two rules by optimizing $O_H^I(\mathbf{a})$ leads to learning at least $\ell \ge 2$ other rules, since $d(R_o, R_s) = 3$, with confidence values greater than $\min(\alpha_o, \alpha_s)$. This means we inevitably learn at least 2 additional **incorrect** rules with substantial confidence values.

Theorem 1 also entails other undesirable issues, for example the resulting list of rules may not have the correct order of importance. More specifically, a rule might have higher confidence value just because it is sharing an atom with another high confidence rule. Thus confidence values are not a direct indicator of rule importance. This reduces the interpretability of the output rules.

We must note that all previous differentiable rule mining methods based on $\Omega_H(\mathbf{a})$ suffer from this limitation. For example Yang et al. [41] has this limitation for rules with maximum length. Section 5.3 illustrates these drawbacks using examples of mined rules.

## 4.2 DRUM

Recall that the number of confidence values for rules of length $T$ or smaller is $(|\mathcal{R}| + 1)^T$. These values can be viewed as entries of a $T$ dimensional tensor where the size of each axis is $|\mathcal{R}| + 1$. To be more specific, we put the confidence value of the rule with body $B_{r_1} \wedge B_{r_2} \wedge \cdots \wedge B_{r_T}$ at position $(r_1, r_2, \ldots, r_T)$ in the tensor and we call it the *confidence value tensor*.

It can be shown that the final confidences obtained by expanding $\Omega_H^I(\mathbf{a})$ are a rank one estimation of the *confidence value tensor*. This interpretation makes the limitation of $\Omega_H^I(\mathbf{a})$ more clear and provides a natural connection to the tensor estimation literature. Since a low-rank approximation (not

just rank one) is a popular method for tensor approximation, we use it to generalize $\Omega_H^I(\mathbf{a})$. The $\Omega$ related to rank $L$ approximation can be formulated as

$$\Omega_H^L(\mathbf{a}, L) \doteq \sum_{j=1}^{L}\{\prod_{i=1}^{T}\sum_{k=0}^{|\mathcal{R}|} a_{j,i,k}\mathbf{A}_{B_k}\}. \tag{7}$$

In the following theorem, we show that $\Omega_H^L(\boldsymbol{a}, L)$ is powerful enough to learn any set of logical rules, without including unrelated ones.

**Theorem 2.** *For any set of rules $R_1, R_2, \cdots R_r$ and their associated confidence values $\alpha_1, \alpha_2, \cdots, \alpha_r$ there exists an $L^*$, and $\boldsymbol{a^*}$, such that:*

$$\Omega_H^L(\boldsymbol{a^*}, L^*) = \alpha_1 R_1 + \alpha_2 R_2 \cdots + \alpha_r R_r.$$

*Proof.* To prove the theorem we will show that one can find a $\mathbf{a}^*$ for $L^* = r$ such that the requirements are met. Without loss of generality, assume $R_j$ (for some $1 \leq j \leq r$) is of length $t_0$ and consists of body atoms $B_{r_1}, B_{r_2}, \cdots, B_{r_{t_0}}$. By setting $a_{j,i,k}^*$

$$a_{j,i,k}^* = \begin{cases} \alpha_j\delta_{r_1}(k) & \text{if } i = 1 \\ \delta_{r_i}(k) & \text{if } 1 < i \leq t_0 \\ \delta_0(k) & \text{if } t_0 < i \end{cases}$$

it is easy to show that $\mathbf{a}^*$ satisfies the condition in Theorem 2. Let's look at $\Omega_H^L(\mathbf{a}^*, L^*)$ for each $j$:

$$\prod_{i=1}^{T}\sum_{k=1}^{|\mathcal{R}|} a_{j,i,k}^*\mathbf{A}_{B_k} = \alpha_j\mathbf{A}_{B_{r_1}} \cdot \mathbf{A}_{B_{r_2}} \cdots \mathbf{A}_{B_{r_{t_0}}} \cdot \mathbf{I} \cdots \mathbf{I} = \alpha_j R_j.$$

Therefore $\Omega_H^L(\boldsymbol{a^*}, L^*) = \sum \alpha_j R_j.$ $\qquad\square$

Note the number of learnable parameters in $\Omega_H^L$ is now $LT(|\mathcal{R}|+1)$. However, this is just the number of free parameters for finding the rules for a single head relation, learning the rules for all relations in knowledge graph requires estimating $LT(|\mathcal{R}| + 1) \cdot |\mathcal{R}|$ parameters, which is $\mathcal{O}(|\mathcal{R}|^2)$ and can be potentially large. Also, the main problem that we haven't addressed yet, is that direct optimization of the objective related to $\Omega_H^L$ learns parameters of rules for different head relations separately, therefore learning one rule can not help in learning others.

Before we explain how RNNs can solve this problem, we would like to draw your attention to the fact that some pairs of relations cannot be followed by each other, or have a very low probability of appearing together. Consider the family knowledge base, where the entities are people and the relations are familial ties like `fatherOf, AuntOf, wifeOf`, etc. If a node in the knowledge graph is `fatherOf`, it cannot be `wife_of` another node because it has to be male. Therefore the relation `wife_of` never follows the relation `father_of`. This kind of information can be useful in estimating logical rules for different head relations and can be shared among them.

To incorporate this observation in our model and to alleviate the mentioned problems, we use $L$ bidirectional RNNs to estimate $a_{j,i,k}$ in equation 7:

$$\mathbf{h}_i^{(j)}, \mathbf{h'}_{T-i+1}^{(j)} = \mathbf{BiRNN}_j(\mathbf{e}_H, \mathbf{h}_{i-1}^{(j)}, \mathbf{h'}_{T-i}^{(j)}), \tag{8}$$

$$[a_{j,i,1}, \cdots, a_{j,i,|\mathcal{R}|+1}] = f_\theta([\mathbf{h}_i^{(j)}, \mathbf{h'}_{T-i+1}^{(j)}]), \tag{9}$$

where $\mathbf{h}$ and $\mathbf{h}'$ are the hidden-states of the forward and backward path RNNs, respectively, both of which are zero initialized. The subindexes of the hidden states denote their time step, and their superindexes identify their bidirectional RNN. $e_H$ is the embedding of the head relation $H$ for which we want to learn a probabilistic logic rule, and $f_\theta$ is a fully connected neural network that generates the coefficients from the hidden states of the RNNs.

We use a bidirectional RNN instead of a normal RNN because it is capable of capturing information about both backward and forward order of which the atoms can appear in the rule. In addition, sharing the same set of recurrent networks for all head predicates (for all $\Omega_H^L$) allows information to be shared from one head predicate to another.

# 5 Experiments

In this section we evaluate DRUM on statistical relation learning and knowledge base completion. We also empirically assess the quality and interpretability of the learned rules.

We implement our method in TensorFlow [1] and train on Tesla K40 GPUs. We use ADAM [19] with learning rate and batch size of 0.001 and 64, respectively. We set both the hidden state dimension and head relation vector size to 128. We did gradient clipping for training the RNNs and used LSTMs [17] for both directions. $f_\theta$ is a single layer fully connected. We followed the convention in the existing literature [41] of splitting the data into three categories of facts, train, and test. The code and the datasets for all the experiments will be publicly available.

## 5.1 Statistical Relation Learning

**Datasets:** Our experiments were conducted on three different datasets [20]. The Unified Medical Language System (UMLS) consists of biomedical concepts such as drug and disease names and relations between them such as diagnosis and treatment. Kinship contains kinship relationships among members of a Central Australian native tribe. The Family data set contains the bloodline relationships between individuals of multiple families. Statistics about each data set are shown in Table 1.

Table 1: Dataset statistics for statistical relation learning

|  | #Triplets | #Relations | #Entities |
|---|---|---|---|
| **Family** | 28356 | 12 | 3007 |
| **UMLS** | 5960 | 46 | 135 |
| **Kinship** | 9587 | 25 | 104 |

We compared DRUM to its state of the art differentiable rule mining alternative, Neural LP [41]. To show the importance of having a rank greater than one in DRUM, we test two versions, DRUM-1 and DRUM-4, with $L = 1$ and $L = 4$ (rank 4), respectively.

To the best of our knowledge, NeuralLP and DRUM are the only scalable [1] and differentiable methods that provide reasoning on KBs without the need to use embeddings of the entities at test time, and provide prediction solely based on the logical rules. Other methods like NTPs [25, 30] and MINERVA [8], rely on *some type of learned* embeddings at training and test time. Since rules are interpretable and embeddings are not, this puts our method and NeuralLP in fully-interpretable category while others do not have this advantage (therefore its not fair to directly compare them with each other). Moreover, methods that rely on embeddings (fully or partially) are prone to having worse results in inductive tasks, as partially shown in the experiment section. Nonetheless we show the results of the other methods in the appendix.

Table 2: Experiment results with maximum rule length 2 and 3

|  |  | **Family** | | | | **UMLS** | | | | **Kinship** | | | |
|---|---|---|---|---|---|---|---|---|---|---|---|---|---|
|  |  |  | Hits@ | | |  | Hits@ | | |  | Hits@ | | |
|  |  | MRR | 10 | 3 | 1 | MRR | 10 | 3 | 1 | MRR | 10 | 3 | 1 |
| $T=2$ | Neural-LP | .91 | .99 | .96 | .86 | .75 | .92 | .86 | .62 | **.62** | .91 | .69 | **.48** |
|  | DRUM-1 | .92 | **1.0** | .98 | .86 | .80 | .97 | .93 | .66 | .51 | .85 | .59 | .34 |
|  | DRUM-4 | .94 | **1.0** | **.99** | .89 | **.81** | **.98** | **.94** | **.67** | .60 | **.92** | .69 | .44 |
| $T=3$ | Neural-LP | .88 | .99 | .95 | .80 | .72 | .93 | .84 | .58 | .61 | .89 | .68 | .46 |
|  | DRUM-1 | .91 | .99 | .96 | .85 | .77 | .96 | .92 | .63 | .57 | .88 | .66 | .43 |
|  | DRUM-4 | **.95** | .99 | .98 | **.91** | .80 | .97 | .92 | .66 | .61 | .91 | **.71** | .46 |

Table 2 shows link prediction results for each dataset in two scenarios with maximum rule length two and three. The results demonstrate that DRUM empirically outperforms Neural-LP in both cases $T = 2, 3$. Moreover it illustrates the importance of having a rank higher than one in estimating confidence values. We can see a more than seven percent improvement on some metrics for UMLS, and meaningful improvements in all other datasets. We believe DRUM's performance over Neural LP

is due to its high rank approximation of rule confidences and its use of bidirectional LSTM to capture forward and backward ordering criteria governing the body relations according to the ontology.

## 5.2 Knowledge Graph Completion

We evaluate our proposed model in inductive and transductive link prediction tasks on two widely used knowledge graphs WordNet [18, 24] and Freebase [3]. WordNet is a knowledge base constructed to produce an intuitively usable dictionary, and Freebase is a growing knowledge base of general facts. In the experiment we use WN18RR [10], a subset of WordNet, and FB15K-237 [36], which both are more challenging versions of WN18 and FB15K [4] respectively. The statistics of these knowledge bases are summarized in Table 3. We also present our results on WN18 [4] in the appendix.

For transductive link prediction we compare DRUM to several state-of-the-art models, including Dist-Mult [40], ComplEx [37], Gaifman [27], TransE [4], ConvE [10], and most importantly Neural-LP. Since NTP($-\lambda$) [30] are not scalable to WN18 or FB15K, we could not present results on larger datasets. Also dILP [14], unlike our method requires negative examples which is hard to obtain under Open World Assumption (OWA) of modern KGs and dILP is memory-expensive as authors admit, which cannot scale to the size of large KGs, thus we can not compare numerical results here.

Table 3: Datasets statistics for Knowledge base completion.

|  | WN18RR | FB15K-237 |
|---|---|---|
| #Train | 86,835 | 272,155 |
| #Valid | 3,034 | 17,535 |
| #Test | 3,134 | 20,466 |
| #Relation | 11 | 237 |
| #Entity | 40,943 | 14,541 |

In this experiment for DRUM we set the rank of the estimator $L = 3$ for both datasets. The results are reported without any hyperparamter tuning. To train the model, we split the training file into facts and new training file with the ratio of three to one. Following the evaluation method in Bordes et al. [4], we use filtered ranking; table 4 summarizes our results.

Table 4: Transductive link prediction results. The results are taken from [21, 41] and [35]

|  | WN18RR | | | | FB15K-237 | | | |
|---|---|---|---|---|---|---|---|---|
|  |  | Hits | | |  | Hits | | |
|  | MRR | @10 | @3 | @1 | MRR | @10 | @3 | @1 |
| R-GCN [31] | – | – | – | – | .248 | .417 | .258 | .153 |
| DistMult [40] | .43 | 49 | .44 | .39 | .241 | .419 | .263 | .155 |
| ConvE [10] | .43 | .52 | .44 | .40 | .325 | .501 | .356 | .237 |
| ComplEx [37] | .44 | .51 | .46 | .41 | .247 | .428 | .275 | .158 |
| TuckER [2] | .470 | .526 | .482 | .443 | .358 | .544 | .394 | .266 |
| ComplEx-N3 [21] | .47 | .54 | – | – | .35 | .54 | – | – |
| RotatE [35] | .476 | .571 | .492 | .428 | .338 | .533 | .375 | .241 |
| Neural LP [41] | .435 | .566 | .434 | .371 | .24 | .362 | – | – |
| MINERVA [8] | .448 | .513 | .456 | .413 | .293 | .456 | .329 | .217 |
| Multi-Hop [23] | .472 | .542 | – | .437 | .393 | .544 | – | .329 |
| DRUM (T=2) | .435 | .568 | .435 | .370 | .250 | .373 | .271 | .187 |
| DRUM (T=3) | .486 | .586 | .513 | .425 | .343 | .516 | .378 | .255 |

The results clearly show DRUM empirically outperforms Neural-LP for all metrics on both datasets. DRUM also achieves state of the art Hit@1, Hit@3 as well as MRR on WN18RR among all methods (including the embedding based ones).

It is important to note that comparing DRUM with embedding based methods solely on accuracy is not a fair comparison, because unlike DRUM they are black-box models that do not provide interpretability. Also, as we will demonstrate next, embedding based methods are not capable of reasoning on previously unseen entities.

For the inductive link prediction experiment, the set of entities in the test and train file need to be disjoint. To force that condition, after randomly selecting a subset of test tuples to be the new test file,

Table 5: Inductive link prediction Hits@10 metrics.

|  | WN18 | FB15K-237 |
|---|---|---|
| TransE | 0.01 | 0.53 |
| Neural LP | 94.49 | 27.97 |
| DRUM | **95.21** | **29.13** |

Table 6: Human assessment of number of consecutive correct rules

| T=2 | Neural LP | DRUM |
|---|---|---|
| father | 2 | 5 |
| sister | 3 | 10 |
| uncle | 6 | 6 |

we omit any tuples from the training file with the entity in the new test file. Table 5 summarizes the inductive results for Hits@10.

It is reasonable to expect a significant drop in the performance of the embedding based methods in the inductive setup. The result of Table 5 clearly shows that fact for the TransE method. The table also demonstrates the superiority of DRUM to Neural LP in the inductive regime. Also for Hits@1 and Hits@3, the results of DRUM are about 1 percent better than NeuralLP and for the TransE all the values are very close to zero.

### 5.3 Quality and Interpretability of the Rules

As stated in Section 1, an important advantage of rules as a reasoning mechanism is their comprehensibility by humans. To evaluate the quality and interpretability of rules mined by DRUM we perform two experiments. Throughout this section we use the *family* dataset for demonstration purposes as it is more tangible. Other datasets like *umls* yield similar results.

We use human annotation to quantitatively assess rule quality of DRUM and Neural LP. For each system and each head predicate, we ask two blind annotators[2] to examine each system's sorted list of rules. The annotators were instructed to identify the first rule they perceive as erroneous. Table 6 depicts the number of correct rules before the system generates an erroneous rule.

The results of this experiment demonstrate that rules mined by DRUM appear to be better sorted and are perceived to be more accurate.

We also sort the rules generated by each system based on their assigned confidences and show the three top rules[3] in Table 7. Logically incorrect rules are highlighted by *italic-red*. This experiment shows two of the three top ranked rules generated by Neural LP are incorrect (for both head predicates $wife$ and $son$).

These errors are inevitable because it can be shown that for rules of maximum length ($T$), the estimator of Neural LP provides a rank one estimator for the *confidence value tensor* described in Section 4.2. Thus according to Theorem 1 the second highest confidence rule generated by Neural LP has to share a body atom with the first rule. For example the rule `brother(B,A)` → `son(B,A)`, even though incorrect, has a high confidence due to sharing the body atom `brother` with the highest confidence rule (first rule). Since DRUM does not have this limitation it can be seen that the same does not happen for rules mined by DRUM.

Table 7: Top 3 rules obtained by each system learned on *family* dataset

| | | | |
|---|---|---|---|
| Neural LP | brother(B, A) ← sister(A, B) | *wife(C, A) ← husband(A, B), husband(B, C)* | son(C, A) ← son(B, A), brother(C, B) |
| | brother(C, A) ← sister(A, B), sister(B, C) | wife(B, A) ← husband(A, B) | *son(B, A) ← brother(B, A)* |
| | brother(C, A) ← brother(A, B), sister(B, C) | *wife(C, A) ← daughter(B, A), husband(B, C)* | *son(C, A) ← son(B, A), mother(B, C)* |
| DRUM | brother(C, A) ← nephew(A, B), uncle(B, C) | wife(A, B) ← husband(B, A) | son(C, A) ← nephew(A, B), brother(B, C) |
| | brother(C, A) ← nephew(A, B), nephew(C, B) | wife(C, A) ← mother(A, B), father(C, B) | son(C, A) ← brother(A, B), mother(C, B) |
| | brother(C, A) ← brother(A, B), sister(B, C) | wife(C, A) ← son(B, A), father(C, B) | son(C, A) ← brother(A, B), daughter(B, C) |

## 6 Conclusion

We present DRUM, a fully differentiable rule mining algorithm which can be used for inductive and interpretable link prediction. We provide intuition about each part of the algorithm and demonstrate its empirical success for a variety of tasks and benchmark datasets.

DRUM's objective function is based on the Open World Assumption of KBs and is trained using only positive examples. As a possible future work we would like to modify DRUM to take advantage of negative sampling. Negative sampling has shown empirical success in representation learning methods and it may also be useful here. Another direction for future work would be to investigate an adequate way of combining differential rule mining with representation learning techniques.

## Acknowledgments

We thank Kazem Shirani for his valuable feedback. We thank Anthony Colas and Sourav Dutta for their help in human assessment of the rules. This work is partially supported by NSF under IIS Award #1526753 and DARPA under Award #FA8750-18-2-0014 (AIDA/GAIA).

## Footnotes

[1] e.g., On the Kinship dataset DRUM takes 1.2 minutes to run vs +8 hours for NTP($-\lambda$) [30] on the same machine.

[2]Two CS students. The annotators are not aware which system produced the rules.

[3]A complete list of top 10 rules is available in the supplementary materials.

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
