[Supplementary Material · drum_appendix.pdf]

# Supplementary Material for DRUM: End-to-End Differentiable Rule Mining on Knowledge Graphs

**Theorem 1.** *If $R_o$, $R_s$ are two rules of length $T$ obtained by optimizing the objective related to $\Omega_H^I$, with confidence values $\alpha_o, \alpha_s$, then there exists $\ell$ rules of length $T$, $R_1, R_2, \cdots, R_\ell$, with confidence values $\alpha_1, \alpha_2, \cdots, \alpha_\ell$ such that:*

$$d(R_o, R_1) = d(R_\ell, R_s) = 1 \quad and \quad d(R_o, R_s) \leq \ell + 1,$$
$$d(R_l, R_{l+1}) = 1 \quad and \quad \alpha_l \geq \min(\alpha_o, \alpha_s) \quad for \quad 1 \leq l \leq \ell,$$

*where $d(.,.)$ is a distance between two rules of the same size defined as the number of mismatched atoms between them.*

*Proof.* Define:
$$\mathbf{a}^* \doteq \arg\max_{\mathbf{a}} O_H^I(\mathbf{a}) = \sum_{(x, H, y) \in KG} \mathbf{v}_x^T \Omega_H^I(\mathbf{a}) \mathbf{v}_y,$$

where $O_H^I(\mathbf{a})$, is the objective related to the $\Omega_H^I(\boldsymbol{a})$ model. The confidence value of a rule of length $T$, for instance $S$, with body $B_{r_1} \wedge B_{r_2} \wedge \cdots \wedge B_{r_T}$, is

$$\alpha_S^* = \prod_{i=1}^{T} a_{i,r_i}^*.$$

Therefore changing a body atom $B_{r_i}$ to $B_{r_i'}$ in $S$, does not decrease the confidence value iff $a_{i,r_i}^* \leq a_{i,r_i'}^*$. Let $a_{i,r_i^*}^*$ be the maximum element of the sequence $a_{i,1}^*, \cdots, a_{i,|\mathcal{R}|}^*$. By consequently changing $B_{r_i}$ in $S$ to $B_{r_i^*}$ (for $i$'s where $r_i^* \neq r_i$) we obtain a sequence of rules of length T, with non-decreasing confidence values. The distance between any two consecutive elements in that sequence is 1. The last element of the sequence ($S^*$) is the rule with the highest confidence value among length $T$ rules, and the length of the sequence is $d(S, S^*) + 1$.

To prove the theorem, it is sufficient to substitute $S$ with $R_o$ and $R_s$ to obtain two sequences of rules, with lengths $d(R_o, S^*) + 1$ and $d(R_s, S^*) + 1$, respectively. by reversing the sequence related to $R_s$ and concatenate it with the other sequence, we have a sequence of length $d(R_o, S^*) + d(R_s, S^*) + 1$, satisfying the conditions required to prove the theorem (after excluding $R_o$ and $R_s$).

The confidences values for the rules in the sequence satisfy the condition $\alpha_l \geq \min(\alpha_o, \alpha_s)$, because all the rules in the sequence related to $R_s$ ($R_o$) and $S^*$ have larger or equal confidence value to $R_s$ ($R_o$). And since $d(.,.)$ is a valid distance function it satisfies the triangle inequality; therefore $d(R_o, R_s) \leq d(R_o, S^*) + d(R_s, S^*)$, which implies $d(R_s, R_o) \leq \ell + 1$. □

Table 1: Comparison with other reasoning methods, an extension to the table 2 in the paper

| Datasets | UMLS | | | | Kinship | | | |
|---|---|---|---|---|---|---|---|---|
| | MRR | Hits@1 | Hits@3 | Hits@10 | MRR | Hits@1 | Hits@3 | Hits@10 |
| ConvE | 0.94 | 0.92 | 0.96 | 0.99 | 0.83 | 0.98 | 0.92 | 0.98 |
| ComplEx | 0.89 | 0.82 | 0.96 | 1 | 0.81 | 0.7 | 0.89 | 0.98 |
| MINERVA | 0.82 | 0.73 | 0.90 | 0.97 | 0.72 | 0.60 | 0.81 | 0.92 |
| NTP[1] | 0.88 | 0.82 | 0.92 | 0.97 | 0.6 | 0.48 | 0.7 | 0.78 |
| NTP-$\lambda$[1] | 0.93 | 0.87 | 0.98 | 1 | 0.8 | 0.76 | 0.82 | 0.89 |
| NTP 2.0 | 0.76 | 0.68 | 0.81 | 0.88 | 0.65 | 0.57 | 0.69 | 0.81 |
| DRUM | 0.81 | 0.67 | 0.94 | 0.98 | 0.61 | 0.46 | 0.71 | 0.91 |

Table 2: Transductive link prediction results

| | **WN18** | | | | **FB15K** | | | |
|---|---|---|---|---|---|---|---|---|
| | | Hits | | | | Hits | | |
| | MRR | @10 | @3 | @1 | MRR | @10 | @3 | @1 |
| DistMult | .822 | .936 | .914 | .728 | .XXX | .XXX | .XXX | XXX |
| ComplEx | .941 | .947 | .936 | .936 | XXX | XXX | XXX | XXX |
| Gaifman | – | .939 | – | .761 | – | – | – | – |
| R-GCN | .814 | .964 | .929 | .697 | XXX | XXX | XXX | XXX |
| TransE | .495 | .943 | .888 | .113 | XXX | XXX | XXX | XXX |
| ConvE | .943 | .956 | .946 | .935 | XXX | XXX | XXX | XXX |
| Neural LP | .94 | .945 | – | – | XXX | XXX | – | – |
| DRUM | .944 | .954 | .943 | .939 | XXX | XXX | XXX | XXX |

Table 3: Transductive link prediction results

| | **WN18** | | | |
|---|---|---|---|---|
| | | Hits | | |
| | MRR | @10 | @3 | @1 |
| DistMult | .822 | .936 | .914 | .728 |
| ComplEx | .941 | .947 | .936 | .936 |
| Gaifman | – | .939 | – | .761 |
| R-GCN | .814 | .964 | .929 | .697 |
| TransE | .495 | .943 | .888 | .113 |
| ConvE | .943 | .956 | .946 | .935 |
| Neural LP | .94 | .945 | – | – |
| DRUM | .944 | .954 | .943 | .939 |

| Head | brother(., .) | wife(., .) | son(., .) |
|---|---|---|---|
| NeuralLP | (B, A) ← inv_sister(B, A)<br>(C, A) ← inv_sister(B, A), inv_sister(C, B)<br>(C, A) ← inv_brother(B, A), inv_sister(C, B)<br>(B, A) ← inv_brother(B, A)<br>(C, A) ← inv_sister(B, A), inv_brother(C, B)<br>(C, A) ← inv_brother(B, A), inv_brother(C, B)<br>(B, A) ← son(B, A)<br>(C, A) ← inv_sister(B, A), son(C, B)<br>N/A<br>N/A | (C, A) ← inv_husband(B, A), inv_husband(C, B)<br>(B, A) ← inv_husband(B, A)<br>(C, A) ← daughter(B, A), inv_husband(C, B)<br>(C, A) ← wife(B, A), inv_husband(C, B)<br>(C, A) ← inv_husband(B, A), mother(C, B)<br>(C, A) ← mother(B, A), inv_husband(C, B)<br>N/A<br>N/A<br>N/A<br>N/A | (C, A) ← son(B, A), brother(C, B)<br>(B, A) ← brother(B, A)<br>(C, A) ← son(B, A), inv_mother(C, B)<br>(C, A) ← inv_mother(B, A), brother(C, B)<br>(B, A) ← inv_mother(B, A)<br>(C, A) ← inv_mother(B, A), inv_mother(C, B)<br>(C, A) ← inv_husband(B, A), brother(C, B)<br>(C, A) ← inv_father(B, A), brother(C, B)<br>(C, A) ← inv_husband(B, A), inv_mother(C, B)<br>(C, A) ← inv_father(B, A), inv_mother(C, B) |
| DRUM | (C, A) ← nephew(A, B), uncle(B, C)<br>(C, A) ← nephew(A, B), inv_nephew(B, C)<br>(C, A) ← brother(A, B), sister(B, C)<br>(C, A) ← brother(A, B), inv_sister(B, C)<br>(C, A) ← brother(A, B), inv_brother(B, C)<br>(C, A) ← brother(A, B), brother(B, C)<br>(C, A) ← nephew(A, B), inv_niece(B, C)<br>(C, A) ← nephew(A, B), aunt(B, C)<br>(C, A) ← inv_uncle(A, B), uncle(B, C)<br>(C, A) ← inv_uncle(A, B), inv_nephew(B, C) | (A, B) ← inv_husband(A, B)<br>(C, A) ← mother(A, B), inv_father(B, C)<br>(C, A) ← inv_son(A, B), inv_father(B, C)<br>(C, A) ← mother(A, B), son(B, C)<br>(C, A) ← inv_son(A, B), son(B, C)<br>(C, A) ← mother(A, B), daughter(B, C)<br>(C, A) ← inv_son(A, B), daughter(B, C)<br>(C, A) ← inv_daughter(A, B), inv_father(B, C)<br>(C, A) ← inv_daughter(A, B), son(B, C)<br>N/A | (C, A) ← nephew(A, B), brother(B, C)<br>(C, A) ← brother(A, B), inv_mother(B, C)<br>(C, A) ← brother(A, B), daughter(B, C)<br>(C, A) ← brother(A, B), son(B, C)<br>(C, A) ← brother(A, B), inv_father(B, C)<br>(C, A) ← inv_sister(A, B), inv_mother(B, C)<br>(C, A) ← inv_sister(A, B), daughter(B, C)<br>(C, A) ← inv_sister(A, B), son(B, C)<br>(C, A) ← inv_sister(A, B), inv_father(B, C)<br>(C, A) ← inv_uncle(A, B), brother(B, C) |

Table 4: Examples of top rules obtained by each system learned on *family* dataset