[Reviews · NeurIPS 2019]

Reviewer 1



Authors propose DRUM, an end-to-end differentiable rule-based inference method which can be used for mining rules via backprop, and extracting rules from data. Their approach is quite interesting - it can be trained from positive examples only, without negative sampling (this is currently a burden for representation learning algorithms targeting knowledge graphs). In DRUM, paths in a knowledge graph are represented by a chain of matrix multiplications (this idea is not especially novel - see [1]). For mining rules, authors start from a formulation of the problem where each rule is associated with a confidence weight, and try to maximise the likelihood of training triples by optimising an end-to-end differentiable objective. However, the space of possible rules (and thus the number of parameters as confidence scores) is massive, so authors propose a way of efficiently approximating the rule scores tensor using with another having a lower rank (Eq. (1)), and by using a recurrent neural network conditioned on the rule. My main concern of this paper is that it does not compare at all with some very relevant work for end-to-end differentiable rule mining, e.g. [2, 3, 4]. Finally, in Tab. 4, authors claim that "The results on FB15k-237 are also very close to SOTA", while from the table it's quite clear they are not (but authors even use bold numbers for their results). Also, in those results, authors evaluate on WN18: this does not seem fair, since [5] shows that one can get WN18 with an extremely simple rule-based baseline, capturing patterns like has_part(X, Y) :- part_of(Y, X). Rules in Table 7 look fairly interpretable, but also do the ones in [2]. Contributions: - Clever end-to-end differentiable rule-based inference method that allows to learn rules via backprop. - Results seem quite promising in comparison with Neural-LP, but there is loads of work in this specific area it would be great to compare with. - Very clearly written paper. [1] https://arxiv.org/abs/1506.01094 [2] https://arxiv.org/pdf/1705.11040 [3] https://arxiv.org/abs/1807.08204 [4] https://arxiv.org/abs/1711.04574 [5] https://arxiv.org/abs/1707.01476

Reviewer 2



- A comparison with neural link prediction methods ComplEx, TransE or ConvE is good, but not timely anymore. I think by now you have to include the following state-of-the-art methods: - M3GM -- Pinter and Eisenstein. Predicting Semantic Relations using Global Graph Properties. EMNLP 2018. - ComplEx-N3 -- Lacroix et al. Canonical Tensor Decomposition for Knowledge Base Completion. 2018. - HypER -- Balazevic et al. Hypernetwork Knowledge Graph Embeddings. 2018. - TuckER -- Balazevic et al. TuckER: Tensor Factorization for Knowledge Graph Completion. 2019. Claims of state-of-the-art performance (L252) do not hold. TuckER, HypER and ComplEx-N3 outperform DRUM on FB15k-237 and all three as well as the inverse model of ConvE outperform DRUM on WN18! Moreover, instead of WN18, I would encourage the authors to use the harder WN18-RR (see Dettmers et al. Convolutional 2d knowledge graph embeddings. AAAI 2018. for details). - It is true that some prior differentiable rule induction work (references [24] and [19] in the paper) was jointly learning entity embeddings and rules. However, at test time one can simply reinitialize entity embeddings randomly so that predictions are solely based on rules. I think that could be a fair and worthwhile comparison to the approach presented here. - How does your approach relate to the random walk for knowledge base population literature? - Wang. (2015). Joint Information Extraction and Reasoning: A Scalable Statistical Relational Learning Approach - Gardner. (2014). Incorporating vector space similarity in random walk inference over knowledge bases. - Gardner. (2013). Improving Learning and Inference in a Large Knowledge-base using Latent Syntactic Cues. - Wang. (2013). Programming with personalized pagerank: a locally groundable first-order probabilistic logic. - Lao. (2012). Reading the web with learned syntactic-semantic inference rules. - Lao. (2011). Random walk inference and learning in a large scale knowledge base. Other comments/questions: - L66: "if each variable in the rule appears at least twice" -- I think you meant if every variable appears in at least two atoms. - L125: What is R in |R|? The set of relations? - L189: What's the rationale behind using L different RNNs instead of one shared one? - L271: How are the two annotators selected? - L292: Extending the approach to incorporate negative sampling should be trivial, no? UPDATE: I thank the authors for their response. I am increasing my score by one.

Reviewer 3



Originality: The paper is an extension of the NeuralLP model wherein changes are made to handle variable length rules. Significance: The paper provides a way to learn variable length rules and improves on the previous by sharing of information while estimating rules for different head relations. Results on the inductive setting for link prediction are promising. The human evaluation is interesting but would benefit from comparison to other methods than just NeuralLP. Clarity & Quality : The paper is well written and does well to explain the problems with simpler approaches and provides solutions in a step-wise manner. However, notations for the equations especially going from page 3 to page 4 should be defined better as the variables used are not adequately described. Issues with the paper: 1) The authors claim that it is not right to compare their method to the black box methods which are not interpretable but also do not report results using models that are, like [1], [2]. These are currently relevant approaches for this task and it is important to report results on them regardless of whether DRUM is able to beat the scores or not. Comparing only to NeuralLP does not seem fair. 2) It has been shown that WN18 is inadequate and recent work on the link prediction task has shifted to using WN18RR instead of WN18 it would be great if the authors reported on this dataset as well. 3) Would like to see hits@1,3 for the inductive setting as well. 4) Equation numbers should be added. [1] Lin, Xi Victoria, Richard Socher, and Caiming Xiong. "Multi-hop knowledge graph reasoning with reward shaping." (2018) [2] Das, Rajarshi, et al. "Go for a walk and arrive at the answer: Reasoning over paths in knowledge bases using reinforcement learning." (2018)

[Author Response · NeurIPS 2019]

We thank all the reviewers for their comments about the novelty and significance of the work. Reviewers all
had constructive suggestions that will improve this paper. Below we address reviewers' two common comments.
3
**Comparing with other baselines and why we mainly emphasize**
**on comparing to NeuralLP:** To the best of our knowledge, Neu-
ralLP and our method are the only scalable[1] and differentiable meth-
ods that provide reasoning on KBs without needing to use embeddings
of the entities at test time, and provide prediction solely based on
the logical rules. Other methods like NTPs [1] and MINERVA [4],
rely on *some type of learned* embeddings at training and test time.
Since rules are interpretable and embeddings are not, this puts our
method and NeuralLP in fully-interpretable category while others
do not have this advantage (therefore its not fair to directly compare
them with each other). Moreover, methods that rely on embeddings
(fully or partially) are prone to having worse results in inductive tasks,
as partially shown in the experiment section. We agree that we didn't
emphasize this point enough and should show their results regardless.
We will **add** the first sentences of this paragraph to our paper. We will
also **remove/clarify** the expression **SOTA** and ambiguous bolding in
tables of experimental results.

Table 1: **Evaluation on harder dataset: WN18RR**
Transductive link prediction results on WN18RR

| WN18RR | MRR | Hits@1 | Hits@3 | Hits@10 |
|---|---|---|---|---|
| ConvE | 0.43 | 0.401 | 0.44 | 0.52 |
| ConvR | 0.475 | 0.443 | 0.489 | 0.537 |
| RotatE | 0.476 | 0.428 | 0.492 | 0.571 |
| TuckER | 0.470 | 0.443 | 0.482 | 0.526 |
| ComplEx-N3 | 0.47 | - | - | 0.54 |
| NTP2.0* (DistMult) | 0.43 | - | - | 0.49 |
| MINERVA | 0.448 | 0.413 | 0.456 | 0.513 |
| Multi-Hop [6] | 0.472 | 0.437 | - | 0.542 |
| Neural LP | 0.435 | 0.371 | 0.434 | 0.566 |
| DRUM, $T=1$ | 0.517 | 0.349 | 0.594 | 0.956 |
| DRUM, $T=2$ | 0.435 | 0.370 | 0.435 | 0.568 |
| DRUM, $T=3$ | 0.486 | 0.425 | 0.513 | 0.586 |

* Results for DistMult, in [2] authors claim NTP 2.0 is on par with a model similar to DistMult.

21
Due to lack of space we briefly address other comments in their order of appearance.

**Reviewer 1:** We really appreciate your thoughtful and detailed comments. Please find in the following our responses.

*Matrix Multiplication Idea:* Thanks for pointing out [5], we will
cite the paper as one of the early works starting the field and write
a brief description. *Comparing with NTP, NTP 2.0, dILP:* We
will add Table 2, and will write a more detailed explanation about
NTP(-$\lambda$) and NTP 2.0 because of their importance. However
since NTP(-$\lambda$) are not scalable to WordNet or FreeBase, we could
not present results on larger datasets. NTP 2.0 does not provide
results on any large datasets, they claim to be on par with a model
similar to distmult which we have added. Thanks for suggesting

Table 2: Comparison with other reasoning methods. Will be appended to Table 2 of paper.

| Datasets | UMLS | | | | Kinship | | | |
|---|---|---|---|---|---|---|---|---|
| | MRR | Hits@1 | Hits@3 | Hits@10 | MRR | Hits@1 | Hits@3 | Hits@10 |
| ConvE | 0.94 | 0.92 | 0.96 | 0.99 | 0.83 | 0.98 | 0.92 | 0.98 |
| ComplEx | 0.89 | 0.82 | 0.96 | 1 | 0.81 | 0.7 | 0.89 | 0.98 |
| MINERVA | 0.82 | 0.73 | 0.90 | 0.97 | 0.72 | 0.60 | 0.81 | 0.92 |
| NTP[1] | 0.88 | 0.82 | 0.92 | 0.97 | 0.6 | 0.48 | 0.7 | 0.78 |
| NTP-$\lambda$[1] | 0.93 | 0.87 | 0.98 | 1 | 0.8 | 0.76 | 0.82 | 0.89 |
| NTP 2.0 | 0.76 | 0.68 | 0.81 | 0.88 | 0.65 | 0.57 | 0.69 | 0.81 |
| DRUM | 0.81 | 0.67 | 0.94 | 0.98 | 0.61 | 0.46 | 0.71 | 0.91 |

dILP [7], we will include it in the references. However, unlike our method [7] requires negative examples which is hard
to obtain under OWA of modern KGs. Also, [7] is memory-expensive as authors admit, and cannot scale to the size of
large KGs (we did not find a publicly available implementation or results on our benchmarks for dILP). *Comparison*
*to relevant work.* please look at the main comment above and we will add a comprehensive comparison table in the
appendix. We will clarify/remove SOTA statements. *Harder data-set evaluation* We will also compare our method's
performance on WN18RR with that of competitors as in Table 1.

**Reviewer 2:** We sincerely appreciate your positive feedback and recognition of the significance of our work.

*More recent work:* We added TuckER, RotatE, and Complex-N3 results for WN18RR. We will also add all of the
other suggested methods to a table in the appendix and add them to the references. *Reinitialize embeddings randomly*:
This is a great idea, since NTP [1] is not scalable and NTP2.0 [2] doesn't provide public code we have to leave this to
future work. *Connections to random walk for KB population literature*: We will summarize these methods and show
connections to DRUM in the final paper. *L66*: You're correct, we modified it in the paper. *L125*: Yes, it is the set of
relations, we defined $\mathcal{R}$ on line 62. *L189*: this is a very thoughtful comment, we tried a shared RNN but the results were
not as good. We believe a single RNN lacks generalizability. *L271*: We asked undergrad CS students. They are not
any of the authors or beneficiaries. *L292*: We considered the OWA of KBs and the effect of wrong negative samples
(actually true but missing) on generating possible "wrong" rules. Though trivial, the cost function and model need
important modifications. Since other methods don't incorporate NS we thought it might not be straight forward.

**Reviewer 4:** We are really thankful for your insightful comments and positive feedback about our work.

*Better notations:* Thanks for the suggestion, we will add more explanation about the notation we used. *Comparing*
*regardless of the results:* The results of Multi-Hop [6] and MINERVA [4] are given in Table 1, we will add a
comprehensive comparison table to the appendix as well. *WN18RR* We agree that we should have included the result,
we will add that to the paper. *hits@1,3* We will add them to the paper, the results for DRUM are about 1 percent better
than NeuralLP and for the TransE all the values are very close to zero. *Equation numbers* we agree that it helps the
readers, we will add them.

**References:** [1] End-to-End Differentiable Proving; [2] Towards Neural Theorem Proving at Scale (NTP 2.0); [3] Traversing
Knowledge Graphs in Vector Space; [4] Go for a Walk and Arrive at the Answer; [5] Traversing KGs in Vector Space; [6] Multi-Hop
Knowledge Graph Reasoning with Reward Shaping; [7] Learning Explanatory Rules from Noisy Data

## Footnotes

[1] e.g., On the Kinship dataset DRUM takes 1.2 minutes to run vs +8 hours for NTP(-$\lambda$) on the same machine.


[Meta-Review · NeurIPS 2019]

This paper proposes an interesting approach for differentiable, interpretable rule mining given a knowledge base. The major pro of the approach is its in an inductive setting without the need for negative examples, which excited the reviewers. Initially the paper lacked important comparisons to many related works, but the author did a good job in rebuttal. Please include the comparison results in the final version and the results on other datasets pointed out by the reviewers. I would like to recommend an acceptance to NeurIPS.